# Development and Up-Scaling of Electrochemical Production and Mild Thermal Reduction of Graphene Oxide

**DOI:** 10.3390/ma15134639

**Published:** 2022-07-01

**Authors:** Markus Ostermann, Peter Velicsanyi, Pierluigi Bilotto, Juergen Schodl, Markus Nadlinger, Guenter Fafilek, Peter A. Lieberzeit, Markus Valtiner

**Affiliations:** 1Centre for Electrochemical Surface Technology, CEST GmbH, A-2700 Wiener Neustadt, Austria; pv@zitt.at (P.V.); juergen.schodl@cest.at (J.S.); markus.nadlinger@cest.at (M.N.); markus.valtiner@cest.at (M.V.); 2Institute of Chemical Technologies and Analytics, Vienna University of Technology, A-1040 Vienna, Austria; guenter.fafilek@tuwien.ac.at; 3Institute of Physical Chemistry, University of Vienna, A-1090 Vienna, Austria; peter.lieberzeit@univie.ac.at; 4Applied Interface Physics, Vienna University of Technology, A-1040 Vienna, Austria

**Keywords:** graphene oxide, reduced graphene oxide, up-scaling, thermal reduction, aeronautical application, polymer filler

## Abstract

To reduce the global emissions of CO_2_, the aviation industry largely relies on new light weight materials, which require multifunctional coatings. Graphene and its derivatives are particularly promising for combining light weight applications with functional coatings. Although they have proven to have outstanding properties, graphene and its precursor graphene oxide (GO) remain far from application at the industrial scale since a comprehensive protocol for mass production is still lacking. In this work, we develop and systematically describe a sustainable up-scaling process for the production of GO based on a three-step electrochemical exfoliation method. Surface characterization techniques (XRD, XPS and Raman) allow the understanding of the fast exfoliation rates obtained, and of high conductivities that are up to four orders of magnitude higher compared to GO produced via the commonly used modified Hummers method. Furthermore, we show that a newly developed mild thermal reduction at 250 °C is sufficient to increase conductivity by another order of magnitude, while limiting energy requirements. The proposed GO powder protocol suggests an up-scaling linear relation between the amount of educt surface and volume of electrolyte. This may support the mass production of GO-based coatings for the aviation industry, and address challenges such as low weight, fire, de-icing and lightning strike protection.

## 1. Introduction

At the end of 2019 the European Union presented the European Green Deal as a counteraction to the ramping climate crisis [1]. The document addressed the challenges for the aviation industry in the CORSIA (Carbon Offsetting and Reduction Scheme for International Aviation) program [2]. In order to reduce their CO_2_ emissions, aviation industries started to explore novel light composite materials. Eventually, it became clear that the composites require fillers and coatings to enhance the adaptability of the aircraft parts to external factors. For instance, while being light, the composite parts need to show resistance to corrosion, to water uptake and to fire, and sufficient conductivity for lightning strike protection. At the same time, it would be ideal to implement thermoelectrical de-icing properties in these composites to avoid the formation of ice clusters on the wing, which may compromise flight safety. Novel materials, e.g., graphene, offer possibilities to obtain such functionalities and preserve the advantage of low composite mass [3].

Graphene is a two-dimensional, atomically thin carbon film, revealing a hexagonal honeycomb structure with sp^2^ binding orbitals. Graphene layers stacked together interact via Van der Waals forces to form graphite. Since its discovery in 2004, graphene has shown outstanding electrical, thermal and mechanical properties, and bio-compatibility. Consequently, many projects aim to implement various graphene-based technologies [4]. On the one hand, fundamental research has proven graphene applications in the fields of nano-electronics and opto-electronics [5], fuel cells [6], energy storage [7], water treatment and decontamination [8,9], lithium-ion batteries [10], super capacitors [11], quantum computers [12], drug delivery [13] and medical applications [14].

On the other hand, research on graphene applied to industrial sectors has found many challenges related to up-scaling and cost in the production of graphene-based materials. In the specific example of the aviation industry, graphene could reduce aircraft weight by 1%, which results in savings of about 1 billion US dollars in terms of reduced fuel consumption and CO_2_ emissions [15]. Moreover, graphene-based coatings with specific electronic features could electrically generate heat and transmit it to the external surface (for de-icing applications), or to enhance conductivity and electromagnetic shielding for lightning strike protection [3,15]. Irrespective of these exciting application potentials, the introduction of large scale graphene-based products into the market remains challenging [3].

Graphene oxide (GO) is a very promising precursor for graphene: it is soluble in water as well as easy to functionalize and to process [16]. GO is mainly synthesized via chemical oxidation of natural graphite, and one of the most used methods was developed by Hummers in 1958. In that work, NaNO_3_ and KMnO_4_ were dissolved in concentrated H_2_SO_4_ to oxidize graphite into graphene oxide flakes [17]. This method was extensively used in the past, but the reaction produces toxic gases such as NO_2_ and N_2_O_4_. [18] On the other hand, modified Hummers method chemical exfoliation can be executed at larger scales. Still, a scaling analysis has not been clarified that might bring to real large scale production [19]. Indeed, chemical production often bears the disadvantage of toxic chemicals and longer reaction time [20], which is why alternative routes are preferential. For instance, improved Hummers methods were presented in the literature [20,21,22] together with alternative green electrochemical multi-step GO production strategies, including ultrasonication [23,24]. The latter often presents an asymmetric electrochemical configuration, where a graphite rod as the working electrode is facing a platinum rod as a counter electrode [25], or alternatively a symmetric configuration with two graphite rods for mass production of GO as inexpensive alternatives. A similar arrangement is used in this work as shown in Figure 1a [26,27,28]. Furthermore, various electrochemical exfoliation methods were investigated with different strategies in terms of intercalation agents aiming to facilitate the process. Sulfate ions, perchlorate ions, alkaline solutions, ionic liquids or even tap water are some application examples [28,29]. Intercalation of species with ionic radii smaller than the graphite interlayer spacing, such as alkali ions, ammonia ions and hydroxide ions, weakens the interlayer bonding of graphite, allowing its structure to swell without major exfoliation during the process [30,31]. Electrochemical exfoliation is assumed to be a reasonable choice for large scale production, but the lack of a standardized upscaling leads to mediocre products/hour rates that can be improved [32,33].

In this paper we provide an upscaling protocol for the production of GO that is, to the best of our knowledge, the highest value per hour presented in the scientific community.

For electrical application, increased conductivity is achieved by reduction of the oxidic groups. Reduction routes such as chemical reduction with hydrazine or ascorbic acid, photocatalytic reduction, or thermal reduction have been reported in the literature [34]. Advantageous due to the easy process setup and high throughput, thermal reduction is often executed at temperatures of 800 °C and higher, with the downside of high energy consumption.

In this work, we further investigate how to produce reduced graphene oxide as a precursor for aeronautical coatings expressing specific conductivity values. For that purpose, it is essential to maintain significant conductive properties, focusing on thermal reduction at lower temperatures for minimal environmental impact. In detail, we present the electrochemical exfoliation of graphene oxide, where anodic pretreatment based on NaOH is exploited to distance the atomic layers in the graphite rods and facilitate exfoliation (Figure 1b). We qualitatively discuss this swelling process and whether the proposed protocol can be promoted to the up-scaling level, highlighting the effective yield. We characterize the produced GO powder with several techniques and study its functionalities. Finally, we thermally reduce the obtained powder to decrease the functional groups and compare the powder conductivity with respect to what is usually produced via the Hummers methods. Our final product maintains its properties throughput different volumina, confirming the upscalability of the process, which quantitatively performed well as it produced 20 g of GO per hour, outperforming the most recent works in this field [32,33].

## 2. Materials and Methods

### 2.1. Materials

All chemicals were used as purchased without further purification. Graphite rods (99% (metals basis)) and KBr (spectroscopy grade, ultrapure) were purchased from Alfa Aesar, NaOH (≥99% ) and KOH (min. 85%) from Carl Roth, H_2_SO_4_ (≥98% Emsure) and Li_2_SO4
·
H_2_O (p.a.) from Merck, Nitrogen (Alphagaz^®^ 99.999%) and Nitrogen with 5 V% Hydrogen addition (Arcal^®^) from Air Liquide, NH_4_OH solution (25%) from VWR, and LiOH
·
H_2_O (ACS reagent, min. 98%) from Fluka.

### 2.2. Electrochemical Graphene Oxide Preparation

The setup for the electrochemical preparation of graphene oxide (GO) is shown in Figure 1a. It consists of a graphite working electrode, two graphite counter electrodes, a power supply, a double-walled reactor, a cooling unit and a mechanical stirrer. The arrangement of the electrodes in an equilateral triangle in the system is shown as a top view in the inset of Figure 1a. Mechanical stirring is positioned in the center of the system to ensure sufficient homogenization and efficient cooling of the electrolyte. The up-scaling protocol is established by testing different volume size reactors (500, 1000 and 1600 mL) with the graphite rods volume increasingly immersed (from 7.9 to 19.0 cm^3^). The ratio of the electrode volume to the reactor volume (see Table 1) is thereby almost constant, resulting in a near-linear up-scaling of the process. In detail, the production protocol went as follow:

First, we treated the working electrode using one of the following electrolytes. Cathodic pretreatment was executed in 1 M LiOH, NaOH, KOH or NH_4_OH or anodic pretreatment in 1 M NaOH. Second, we applied an anodic exfoliation step in 1 M H_2_SO_4_ on the electrode. An additional exfoliation experiment in 1 M H_2_SO_4_ and 0.1 M Li_2_SO_4_ was executed to evaluate the effect of adding Li^+^ to the exfoliation electrolyte. Third, we moved the product to an ultrasonication bath for 2 h to further promote the exfoliation. Fourth, we performed vacuum filtration followed by washing with deionized water to remove electrolyte residuals. Fifth, we dried the resulting powder at 50 °C and 10 mbar for 24 h before further processing. Figure 1b summarizes the process steps with an anodic pretreatment in 1 M NaOH, subsequent exfoliation and ultrasonication post-treatment.

### 2.3. Thermal Reduction of Graphene Oxide

For the thermal reduction of 1 g GO powder, we positioned the material in a quartz boat and placed it in a push type tube furnace. An inert (N_2_) or reducing (N_2_ + 5 V% H_2_) atmosphere was set by passing gas through the furnace. We applied a heating rate of 20 °C/min to reach the targeted reduction temperature. The temperature was kept constant for the targeted reduction time under gas flow. We further analyzed the powder once the furnace came back to room temperature. To determine the effects of temperature, time and atmosphere on the resulting reduced graphene oxide (rGO) powder, we utilized the software Design expert^®^ 12 by Stat-Ease^®^ to establish a two-factorial model with single repetition and two center points and calculated the influence of each parameter in play. Table 2 summarizes the chosen design limits for each parameter (reduction temperature *T*_R_, time *t* and H_2_ addition *V*_H_2__). To prevent possible deviations due to the educt material, we used powder produced from the same exfoliation batch.

### 2.4. Powder Conductivity Measurement

As a starting point, we grated the obtained powders to avoid any agglomerations influencing the conductivity measurement. Then, we placed a teflon matrix on a copper base and a 100 mg powder sample into the matrix. We used an additional copper piston to confine the powder sample from the top. We applied a load of 500 N on the piston, corresponding to a pressure of 4.42 MPa. The pressure was controlled with a Sauter FH 500 and a Sauter wheel test manual stand. We recorded the resistance of the powder sample under pressure using a micro-ohmmeter (ndp technologies DRM-10A) with the four-point method connected to the copper pistons. We measured the pellet thickness via the dissection of the piston. The resistance of the copper matrix was not influencing the overall resistance, which was confirmed by blank measurements showing a resistance value lower by at least two orders of magnitude. Then, we polished the contacts of the copper matrix after each measurement to avoid systematic errors due to an oxidic layer between the sample and the matrix. The measurement of 
σ
 was executed three times per sample to ensure reproducibility.

The powder conductivity 
$\sigma_{Pellet}$
 was calculated using Equation (Equation 1), where 
$d_{Pellet}$
 describes the pellet thickness under pressure, 
$A_{Pellet}$
 complies with the area of the pellet (=1.13 cm^2^) and *R* is the measured resistance.

(1)
$$\sigma_{powder}=\frac{d_{Pellet}}{A_{Pellet}}\;\cdot \;\frac{1}{R}\;\left[S/m\right]$$


A similar test strategy for measurements of the conductivity of carbonaceous powders was already reported by Celzard et al. [35] and Marinho et al. [36].

### 2.5. X-ray Diffraction (XRD)

We performed X-ray diffraction (XRD) employing a PANanalytical Empyrean set-up (Malvern Pananalytical Ltd., Malvern, UK) to determine the crystal structure, number of layers and purity of the produced powder. We prepared the dried powder on a Si wafer measuring in the range 5–90°. To calculate the distribution of *n* in the product, the (002) reflex was fitted by three gaussian functions to determine the amount of few-layered (<10 layers), multi-layered (10–25 layers) and graphitic (>25 layers) parts in the powder. Using the fit data, the interlayer space was calculated via Bragg’s law (Equation (Equation 2)) with the diffraction angle 
θ
 and the wavelength of the X-ray 
λ
 (0.15406 nm). Using Scherrer’s equation (Equation (Equation 3)) crystallite size in z-direction was calculated using the full width at half maximum 
β
 of the (002) reflex fitted curves, the Scherrer form factor *K* (0.89) and the wavelength of the X-ray 
λ
 (0.15406 nm). This resulted in an interlayer spacing *d* (nm) and a crystallite thickness *T* (nm) for the parts of the graphene oxide powder. By dividing *T* by the interlayer space *d* added to the thickness of an atomic layer *a* (0.1 nm), the approximate number of layers was calculated by Equation (Equation 4). The area fitted from the gaussian curves corresponds to the amount of few-layered (<10 layers), multi-layered (10–25 layers) and graphitic (>25 layers) parts in the powder.

Similar approaches for the determination of various graphene oxide powders were already described by Huh [37], Andonovic et al. [38] and Sharma et al. [39].

(2)
$$d=\frac{\lambda }{2\;\cdot \;\sin\theta }\;\left[\mathrm{nm}\right]$$


(3)
$$T=\frac{K\;\cdot \;\lambda }{\beta \;\cdot \;\cos\theta }\;\left[\mathrm{nm}\right]$$


(4)
$$n=\frac{T}{d+a}$$


### 2.6. Characterization

Additionally to the XRD measurements mentioned above, materials were characterized via cyclic voltammetry (CV), Raman spectrometry (Raman), X-ray photoelectron spectroscopy (XPS) and infrared spectroscopy (IR). We recorded CV data with a Biologic SP-240 potentiostat (Bio-Logic Science Instruments Ltd, Seyssinet-Pariset, France) in three electrode cell configuration to determine electrochemical reactions of the graphite educt material. Specifically, we used Ag/AgCl in 3 M KCl for measurements in NaOH or Hg/HgSO_4_ in saturated K_2_SO_4_ for measurements in H_2_SO_4_ as reference electrodes. We utilized a confocal Raman spectrometer with a 50 µm confocal pinhole (DXRxi from Thermo Fisher Scientific Ltd, Loughborough, UK) to record Raman spectra in the range of 50–3400 cm^−1^ of GO powders and graphite educt to evaluate the respective defects in the powders. During the measurements, we applied an intensity of 7 mW for 532 nm green laser, and an integration time of 4000 ms and 15 scans averaged per measurement. We executed XPS measurements with a Thetaprobe XPS system from Thermo Scientific (Thermo Fisher Scientific Ltd, Loughborough, UK) to determine oxygen content and ratio of oxygen containing groups in the GO product. Finally, we measured IR-spectra on a Tensor 27 Hyperion (Bruker Corporation, Billerica, United States) using the KBr pellet method to determine the presence of functional groups in the powder.

## 3. Results and Discussion

First, we verified the upscaling using three increasing batch sizes with linearly scaling electrode volume to reactor volume ratios. We checked the product quality with electrochemical measurements of the graphite educt. Second, we characterized the layer distribution, the defect density of the GO powder, and the amount of oxidic groups using XRD, Raman spectroscopy and XPS, respectively, and we assessed their potential to serve as in-line quality controls of an industrial scale production process. Third, we evaluated the powder conductivity to show the advantages of the mild electrochemical oxidation compared to typically applied (modified) Hummers methods. Finally, we applied subsequent thermal treatment to further reduce the powder and restore the aromatic backbone, improving the conductivity.

### 3.1. Cyclic Voltammetry

Figure 1c,d shows cyclic voltammetric measurements of graphite educt rods in 1 M NaOH and 1 M H_2_SO_4_. Both measurements led to a visible evolution of hydrogen on the cathodic side and oxygen on the anodic side due to the electrolysis of water. In 1 M NaOH a hysteresis appeared in the first scan on the anodic side between 1 and 2 V, as shown in the inset in Figure 1c. This corresponds to the intercalation of hydroxyl anions into the graphite lattice resulting in a swelling of the graphite rod. This intercalation weakens the interactions between the graphite layers, but does not lead to exfoliation. The absence of major peaks in the negative scan direction indicates only minor oxidation of the graphite rod during treatment in 1 M NaOH. Besides intercalation, the electrode surface was roughened due to the formation of oxygen in the positive scan direction. Loose particles on the surface were removed resulting in a bigger active surface area and increased currents during the subsequent cycles. Figure 1d presents the CV measurement of an untreated graphite rod (in red) and an anodically pretreated graphite rod (in blue; further details on the pretreatment in the material section) in 1 M H_2_SO_4_ as the electrolyte. One can observe two peaks between −1 and 2 V. They are attributed to the reduction of oxidic groups previously formed in the positive scan direction and intercalated sulfate ions being released again in the negative scan direction. These peaks are prominent in the pretreated sample indicating more oxidation and intercalation in the positive scan direction. Pretreatment led to the removal of loosely bound particles from the surface due to the evolution of oxygen on the working electrode, resulting in a surface roughening (see detailed picture in Appendix A).This minor amount of particles is visible in the pretreatment electrolyte as a sediment. Roughening produced a more accessible active area for electrolyte oxidation during the positive scan direction and improved the interaction between graphite and electrolyte, leading to higher currents.

### 3.2. XRD and Raman Analysis

Figure 2a compares XRD measurements on the graphite educt (red) and GO produced with anodic pretreatment in 1 M NaOH during up-scaling at the reactor volumes of 500 mL (blue), 1000 mL (yellow) and 1600 mL (green). The most prominent peak in the diffractogram is the (002) reflex at 26.62°. This peak position indicates a predominant low oxidation degree of the product. [20] The educt graphite shows a very sharp peak structure indicating crystallites comprising a vast number of layers. The inset in Figure 2a displays an asymmetric profile in the (002) reflex. The latter indicates a distribution of different layer numbers *n* present in the powder due to peak broadening with lower layer numbers [38]. Increased interlayer spacing after the exfoliation and oxidation process caused the slight shift in maximum for both few-layered and multi-layered powder parts to lower diffraction angles leading to asymmetry in the lower diffraction angle direction. The peak structure and its asymmetric profile are in accordance with the literature [38].

For comparison, we also performed the same analysis on powders with different cathodic and anodic pretreatment steps and Li^+^ addition to the exfoliation electrolyte to test ion-specific effects. (Data shown in Appendix A). All powders showed a similar broadened reflex structure compared to the graphite educt, again confirming the importance of hydroxide intercalation. Interestingly, the powder with the Li_2_SO_4_ addition appears to have a narrower (002) reflex suggesting that the co-ion also plays a role during exfoliation. As such, the choice of the co-ion may offer a future optimization path.

To further elucidate the nature of our GO powder, we applied Gaussian fit analysis to calculate the number of layers *n*, as discussed in the methods section. Figure 2b shows the distribution of different layer numbers *n* in the graphite educt and produced graphene oxides with anodic pretreatment in 1 M NaOH solution during up-scaling. The educt consists solely of materials comprising more than 25 layers. The electrochemical graphene oxide shows a distribution of different layer numbers as follows: About 60% of the fitted XRD data show 
n<10
 for each up-scaling process, confirming a positive performance of the GO powder comprising only few layers. The other parts are: 23% of multi-layered grains with *n* from 10 to 25 layers, and graphitic residues are similar to the educt material. The latter, to our understanding, may be related to exfoliation errors such as loss of electrical contact during polarization (e.g., flake off) [29,32]. Finally, the data show that the number of layers *n* does not vary when using different electrolytes, except when adding Li_2_SO_4_ to the exfoliation electrolyte, as previously mentioned. In that case, about 70% of the material revealed 
n>25
 (data shown in Appendix A).

Figure 2c shows the Raman spectra of the graphitic educt (red) and the GO powders during up-scaling at the following volumes: 500 mL (blue), 1000 mL (yellow) and 1600 mL (green). The educt shows the D band at 1350 cm^−1^, the G band at 1576 cm^−1^, the D’ band at 1615 cm^−1^ and the 2D band at 2700 cm^−1^. The GO powders also show these bands with additional interbands in the first-order region (1100–1800 cm^−1^). These defect-induced bands are the D* band at 1150–1200 cm^−1^, the D” band at 1510 cm^−1^ and the D’ band at 1615 cm^−1^. In the second-order region (2400–3300 cm^−1^) also additional bands beside the 2D band appeared after the exfoliation. These are the G* band at 2490 cm^−1^, the D + D’ band at 2940 cm^−1^ and the 2D’ band at 3200 cm^−1^. Exfoliation led to activation of first-order defect bands and increased the intensity of the D band. Additionally, one can observe a slight shift of the G band to higher wave numbers (1585 cm^−1^). These effects are attributed to introducing oxidic groups into the aromatic graphite lattice [40]. In the second-order region the intensity of the 2D band significantly decreased due to exfoliation; thus, other overtones appeared [40,41]. To further investigate the Raman spectra, the first-order region was fitted according to López-Díaz et al. [40] with two Gaussian functions (D* and D”) and three Voigt functions (D, G, D’). The fitted data were used to calculate the intensities *I* of the defect bands normalized to the G band. Figure 2d summarizes the results. The graphite educt showed an 
$ID/I_{G}$
 ratio of about 0.2 and additionally an 
$ID^{\prime }/I_{G}$
 ratio of about 0.05. The other defect bands were not activated, as the powder was not oxidized yet. On the contrary, the produced GO powder during up-scaling exhibited an 
$ID/I_{G}$
 ratio of about 1.25–1.35, as well as an 
$ID*/I_{G}$
 ratio of about 0.03–0.04, an 
$ID^{\prime \prime }/I_{G}$
 ratio of about 0.14–0.2 and an 
$ID^{\prime }/I_{G}$
 ratio of about 0.37–0.46. These ratios are comparable during linear up-scaling, indicating no significant deviation in product properties as a function of larger volumes. The ratios of the defect bands are lower compared to GO powders produced by chemical exfoliation methods, [40] indicating lower oxidation of the graphite during exfoliation resulting in superior electrical properties. Moreover, the Raman data highlight that the quality of our up-scalable GO powder is in the range of investigated commercially available powders [42].

Measuring graphene oxide produced in other electrolytes (Appendix A)), shown in Appendix A, reveals an 
$ID/I_{G}$
 ratio ranging from 1.15 to 1.5. Only the powder produced through cathodic pretreatment in 1 M NH_4_OH shows significantly lower ratios attributed to different functionalization (e.g., ammonia groups) or a cation-related effect during that process, which again demonstrates that anodic OH^-^ intercalation is essential. Additionally to the Raman spectra, the electrical conductivities 
σ
 of these powders were measured (Appendix A). Powders produced without pretreatment and with anodic pretreatment in 1 M NaOH exhibited better conductivity than the others. We attribute this behavior to the weaker deterioration of the conductive aromatic network in these samples by oxidation. The conductivity of the powder produced without pretreatment was highest and more comparable to the educt due to lower defect density. The benefits of anodic pretreatment are activation of the surface, and the resulting faster kinetics of the exfoliation. It approximately takes 20–30% less time to exfoliate the same amount of materials after anodic pretreatment.

### 3.3. Electrical Conductivity and XPS Measurements

Figure 3a displays XPS spectra of GO powder produced in a 1600 mL reactor (batch I). The spectra exhibits a complex C1s peak (292–284 eV) with multiple peaks and a broad O1s peak (533 eV), which can be fit with a single component. We further analyzed the respective oxygen functional groups on carbon by fitting the C1s spectra according to the literature [20,43]. Good fitting was achieved with six gaussian functions, indicating C-C/C-H, C-OH, C-O-C, O-C-O/C=O, O=C-O and pi-pi* carbon species. Figure 3b summarizes the deconvoluted XPS data of a graphene oxide powder produced in a 500 mL reactor and three graphene oxide powders produced in a 1600 mL reactor. Appendix A depicts the fitted data of the respective C1s peaks for all materials.

In detail, the XPS measurements showed a C/O ratio of about 4/1 (20 at%). The low oxidation degree corresponds to the position of the (002) diffraction reflex at about 26°. The differences in oxygen content during up-scaling of the process appear negligible and are attributed to minor deviations in other process parameters. The Pi-Pi* shake-up at 291.3 eV confirms the presence of a conductive aromatic backbone with double-bonded sp^2^ carbon being the major species in the powder. This is in line with the mild oxidation and reasonable conductivity of the product. The remaining carbon appears in different oxidic functional groups with the majority related to ether groups (≈18 at%), indicating a functionalization of the aromatic graphene backbone.

Beside the ether groups, also hydroxylic groups (≈8 at%) situated in-plane and at the edges, and carbonylic (≈6 at%) and carboxylic groups (≈4 at%) situated at the edges were present in the powder, as shown in Figure 3.

Overall the dominating presence of ether groups indicates higher in-plane functionalization compared to edge functionalization. Pretreatment generates oxidic groups through the reaction of graphite with intercalating hydroxide ions leading to swelling. Subsequently, graphite is oxidized out during exfoliation due to the reaction with sulfuric acid and oxygen formed at the working electrode. On the one hand, the oxidic groups’ presence can reduce the aromaticity, thus impacting powder conductivity. On the other hand, they prevent restacking of layers and offer the possibility of further chemical functionalization. Compared to GO powders produced via chemical exfoliation (e.g., modified Hummers method) with an oxygen content of 30–50 at%, the presented electrochemical protocol achieves milder oxidation and therefore, lower oxygen content of about 20 at%.

In Table 3 we report the conductivity 
σ
 (S/m) for the graphite educt and six different batches produced according to the final up-scaled protocol in the 1600 mL reactor with anodic pretreatment in 1 M NaOH. Due to oxidation during the exfoliation process, the conductivity of the resulting graphene oxide powder is about an order of magnitude lower than in the graphite educt.

This is attributed to defects introduced via oxidic groups and ultrasonication. Minor deviations of powder conductivity relate to fluctuations in reactor temperature and variations of the educt material. The final value obtained with the proposed method is notably higher by about four orders of magnitude compared to powder produced via the modified Hummers method (1.56 × 10^−2^ S/m) [44] confirming the viability of our up-scaling protocol. As a comparison, e.g., Marinho et al. [36] reported powder conductivity (2.62 × 10^2^ S/m), similar to our product, for a commercial graphene powder, produced by harsh chemical graphite oxidation and subsequent thermal exfoliation [45], with 15 at% and a single-layer content of at least 50%. This highlights the commercial viability and sustainability of this newly proposed strategy based on electrochemical exfoliation and thermal post-treatment involving environmentally benign chemicals.

### 3.4. Thermal Reduction

Although the conductivity measured is higher than the one from modified Hummer methods, a further increment is demanded for electricity-based aeronautical applications such as de-icing and lightning strike protection. To achieve higher electrical conductivity, we proceeded with thermal reduction (TR) to generate reduced graphene oxide (rGO). In the ideal case, full reduction would remove all functional groups from the graphene layer, as represented in the Appendix A. In reality, it is not easy to obtain a completely reduced layer. Our strategy is to apply thermal annealing in inert (N_2_) or a reducing (N_2_ + H_2_) atmosphere to induce removal of oxidic groups and increase conductivity by restoring aromatic systems.

A two-factor model varying reduction temperature 
$T_{R}$
, time *t* and H_2_ addition 
VH2
 was applied to investigate the influence on the resulting powder conductivity 
σ
 (see the materials section for further details). Appendix A summarizes the runs and respective 
σ
. A Pareto-chart (Appendix A) suggests that 
$T_{R}$
 is the only parameter having significant influence on the resulting powder conductivity. Reduction time and H_2_ addition to the process gas, as well as parameter combination factors are below the significance limit (t-value) and therefore irrelevant within the design limits. This indicates that adding hydrogen does not lead to increased reduction of functional groups or improved restoration of the aromatic backbone. This is in line with XPS data and suggests that in-plane ether groups can be split thermally from the material, while the edge groups are likely stronger bound.

To understand the reduction mechanism in more detail, we further varied the temperature (Run 21–26; Appendix A) to verify the dependence on 
$T_{R}$
. The goal was finding a mild temperature window where powder conductivity is in a range that is of interest in industrial applications, while reducing the energy consumption.

Figure 4 displays powder conductivity and mass loss of GO as a function of the reduction temperature 
$T_{R}$
. While the temperature increases, GO is reduced and functional groups start to decompose, enhancing powder conductivity. In detail, the first jump in 
σ
 appears at about 200–250 °C after reduction of about half of the functional groups (about 12.5 w%). This indicates that decomposition of functional groups (e.g., ethers) restores the aromatic backbone and significantly elevates conductivity. Reduction until 600 °C results in a smaller increase in conductivity despite decomposition of an additional 5 w%. A further increase of the reduction temperature results in a second step in conductivity. The increase in mass loss during the final step is smaller (about 2.5 w%). It corresponds to the reduction of the last functional groups deteriorating the conductivity of the powder, achieving almost full reduction at 20 w% mass loss. Hence, in summary, the thermal reduction proceeds initially via ether groups, and then it continues via edge groups with increasing temperature. The overall mass loss is in full agreement with the oxygen content measured with XPS.

Reference measurements of 
σ
 after wet chemical ascorbic acid reduction (1.47 × 10^3^ S/m) and hydroiodic acid reduction (2.11 × 10^3^ S/m) indicate similar conductivities compared to the thermal reduction protocol. This suggests that thermal reduction may be a viable alternative to wet chemical reduction.

Electrical conductivity after thermal reduction of GO produced via a modified Hummer’s method showed initially lower conductivity after reduction at similar temperatures by a factor of at least 2–3 during a mild reduction at 250–300 °C. A further increase of the reduction temperature to 600–700 °C is necessary to receive similar conductivity with chemically exfoliated starting material. With reduction temperatures of 800 °C corresponding to a near full reduction, the conductivity values strongly resemble each other [46,47]. This shows a clear advantage of our electrochemical starting material with a lower degree of oxidation compared to GO produced using Hummer’s method.

We further selected samples of rGO powders at different 
$T_{R}$
 and characterized them with Raman and IR spectroscopy. First, for the Raman spectra, we analyzed three different spots per sample. The first-order region (1000–1800 cm^−1^) was fitted as mentioned above according to López-Díaz et al. [40] with two Gaussian functions (D* and D”) and three Voigt functions (D, G, D’) to deconvolute the defect bands and the G band. The resulting intensity ratios to the G band (Appendix A) show a steady decrease of defected band intensity. The 
$ID/I_{G}$
 value is stable at about 1.35–1.45, until the last reduction step at 800 °C, where a second strong increase in conductivity and a decrease in the D band intensity occurs. Thus, the highly oxidized groups attributed to the D band decompose at temperatures between 600 and 800 °C. The intensities of the D’ and D” bands show similar behavior during thermal reduction. The finding confirms that conductivity increases due to oxidic groups’ decomposition, causing an increase in crystallinity of the material, which results in weaker defect band intensity. By contrast, the intensity of the D* band remains stable during the reduction process. Second, we recorded IR spectra of rGO samples after seven different reduction temperatures (Appendix A). The electrochemically produced educt GO powder showed significant IR bands at 3434 cm^−1^ (O–H stretching), 1723 cm^−1^ (C=O stretching), 1578 cm^−1^ (C=C stretching), 1385 cm^−1^ (C–O stretching of carboxylic group), 1214 cm^−1^ (C–O–C stretching) and 1124 cm^−1^ (C–OH stretching) [48,49].

These bands are in agreement with XPS results indicating the presence of ether, hydroxyl, carbonylic and carboxyl groups in the educt powder. Until a reduction temperature of 250 °C, one cannot observe any major differences in the IR spectra. Band intensities related to oxygen-containing groups steadily decrease with higher reduction temperatures. After reduction at 800 °C the bands of the O–H stretching vibration (3434 cm^−1^), C=C stretching vibration (1578 cm^−1^) and C–O–C stretching vibration (1214 cm^−1^) are observed indicating a nearly complete decomposition of the oxidic groups.

These results confirm that oxidic groups decompose gradually during thermal annealing, in agreement with Raman spectroscopy and the mass loss during reduction. Although some defects are still present, the conductive aromatic backbone basically gets restored.

## 4. Conclusions

In the present work we investigated a new protocol to produce graphene oxide (GO) powder for industrial up-scaling in a sustainable manner. We tested the method for three different volume to area ratios for electrolyte and graphite precursors. Our data indicate near linear scaling, confirming that this method allows for cheap large scale production up to the kg level. Powder characterization methods (XRD, XPS, Raman) revealed consistent features as a function of the production scale. XRD measurements supported that about 60 % of the powder presents less than ten layers, while XPS data indicated that the oxygen content in the produced GO powder is lower compared to products obtained via (modified) Hummers methods, preserving four order of magnitude higher conductivity. XRD and Raman are viable options for in-line quality measurement systems regarding an industrial application of the proposed protocol.

Furthermore, the obtained conductivity 
σ
 could be enhanced by a factor of 4 after mild thermal reduction at 
TR=250°C
, saving considerable energy compared to classical reduced GO production processes. The proposed strategy thus paves the way to various industrial applications of graphene-based materials for future usage where the cost and environmental impact are design parameters. As an outlook, we aim to explore our GO powder for the next generation of multifunctional coatings in the aviation industry, including de-icing, lightning strike protection, corrosion resistance and/or water uptake. 

## Figures and Tables

**Figure 1 materials-15-04639-f001:**
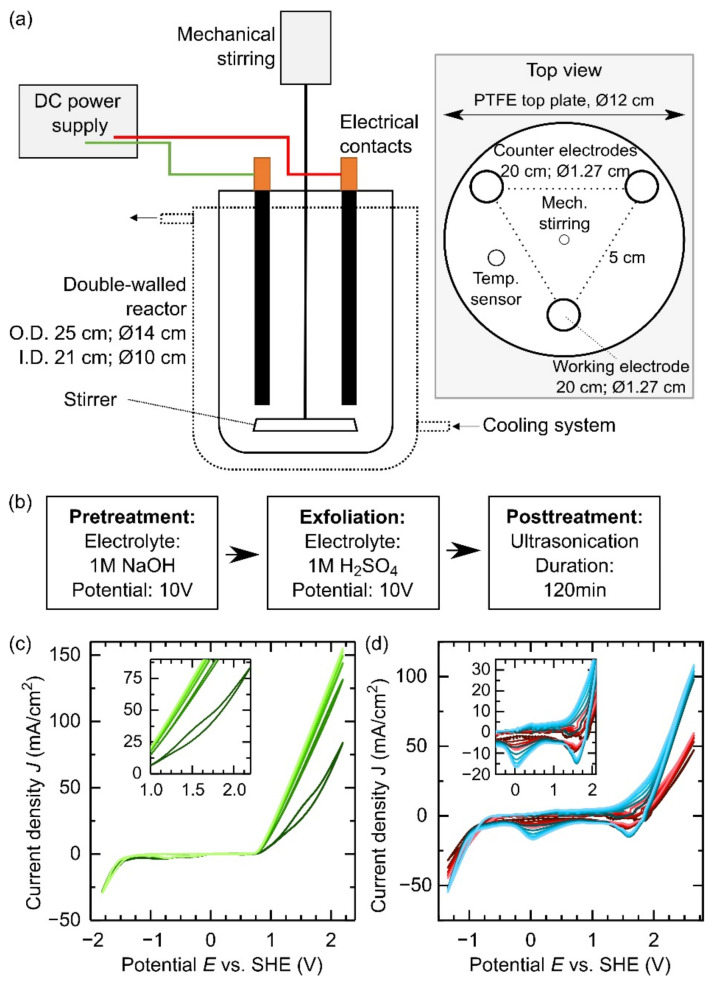
(**a**) Front (**left**) and top (**right**) sides of the electrochemical exfoliation set-up. The lateral view shows the cooling system and the power supply while the top view pictures the electrodes’ position in the electrochemical reactor. The dimensions refer to a 1600 mL reactor. (**b**) Production protocol for the up-scaling process with anodic pretreatment in 1 M NaOH. (**c**) Cyclic voltammetry applied to a graphite rod in 1 M NaOH (scan rate 10 mV/s; 6 cycles: dark green to light green). (**d**) Cyclic voltammetry applied to a graphite rod in 1 M H_2_SO_4_ (scan rate 10 mV/s; 6 cycles: dark color to light color). Red indicates an untreated graphite rod and blue indicates a graphite rod pretreated anodically in 1 M NaOH for 10 min.

**Figure 2 materials-15-04639-f002:**
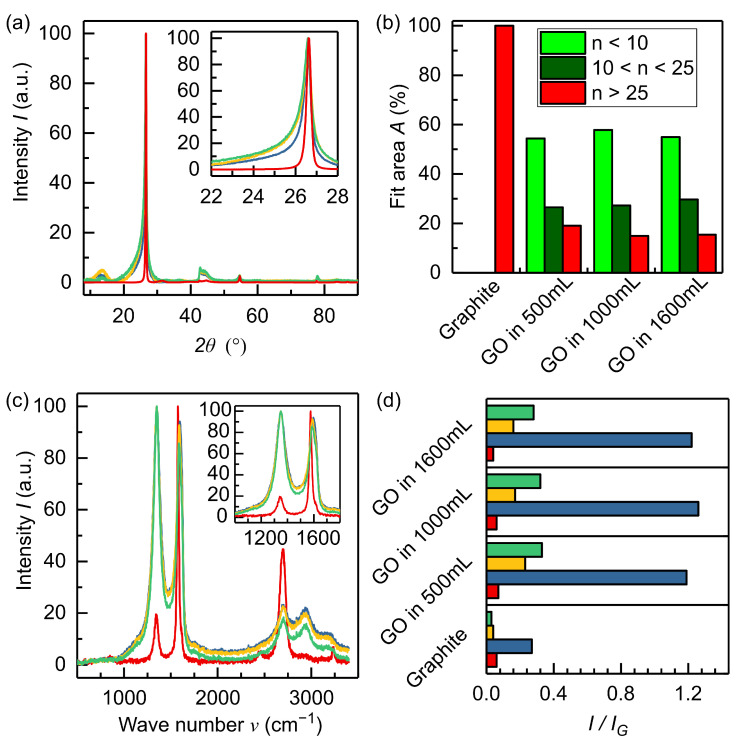
(**a**) XRD diffractogram of graphite educt (red) and electrochemically produced graphene oxide powders at different up-scaling stages (500 mL reactor (blue), 1000 mL reactor (yellow) and 1600 mL reactor (green)); (**b**) distribution of layer numbers calculated from the (002) reflex of the respective XRD measurement; (**c**) Raman spectra of graphite educt (red) and graphene oxide powders at different up-scaling stages (500 mL reactor (blue), 1000 mL reactor (yellow) and 1600 mL reactor (green)); (**d**) calculated Intensity ratio to the G band 
$I/I_{G}$
 of first-order defect bands (D* red, D blue, D” yellow, D’ green).

**Figure 3 materials-15-04639-f003:**
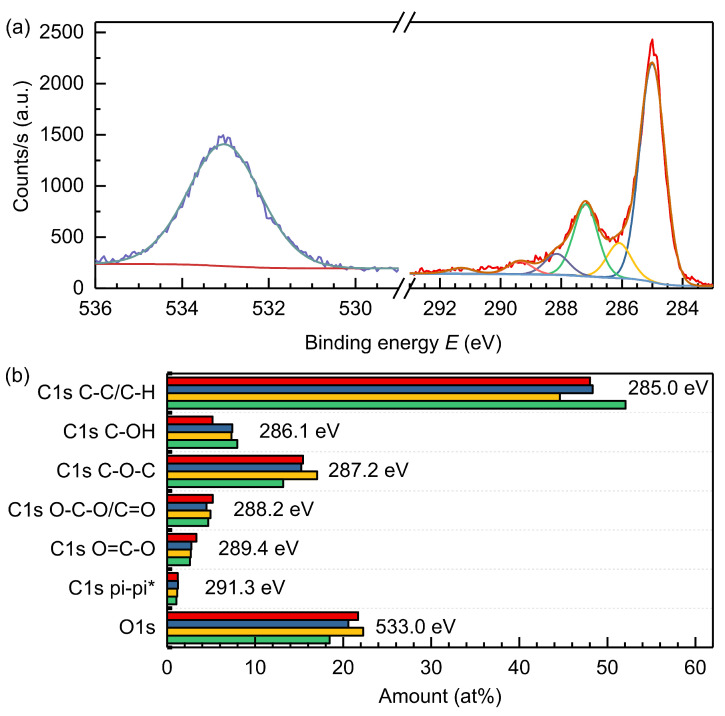
(**a**) XPS spectra of the GO powder produced in a 1600 mL reactor (Batch I) with the deconvolution of the C1s peak (raw data (red), C-C/C-H (blue), C-OH (yellow), C-O-C (green), O-C-O/C=O (violet), O=C-O (grey), pi-pi* (salmon)) and O1s peak (raw data (dark blue), fit (turquoise)); (**b**) deconvoluted XPS data showing the amount of functional groups in electrochemically produced graphene oxide powders in a 500 mL reactor (red) and three different batches produced in a 1600 mL reactor (Batch I blue, Batch II yellow, Batch III green).

**Figure 4 materials-15-04639-f004:**
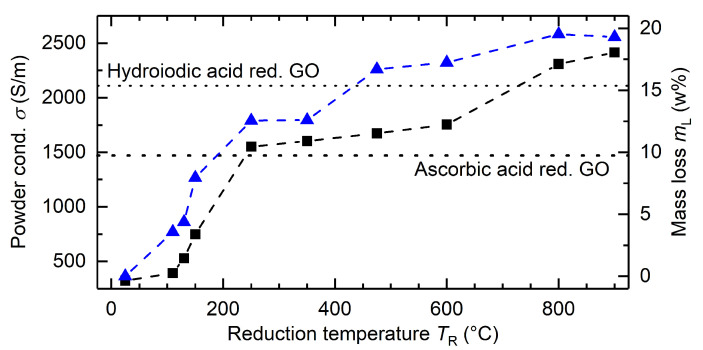
The graph shows on the left axis (in black) the powder conductivity 
σ
 and on the right axis (in blue) the reduction mass loss 
$m_{L}$
, both as a function of the reduction temperature *T*_R_.

**Table 1 materials-15-04639-t001:** Reactor volume, immersed working electrode volume and electrolyte volume to graphite working electrode volume ratio during the three different stages of up-scaling.

Reactor Volume [mL]	Immersed Working Electrode Volume [cm^3^]	Electrode Volume/Reactor Volume [cm^3^/L]
500	7.9	13.9
1000	12.7	12.7
1600	19.0	11.9

**Table 2 materials-15-04639-t002:** Parameter limits of the thermal reduction 2-factorial screening design.

Parameter	Lower Limit	Upper Limit
Reduction temperature *T*_R_ [°C]	350	600
Time *t* [min]	30	180
H_2_ addition *V*_H_2__ [V%]	0	5

**Table 3 materials-15-04639-t003:** Powder conductivity 
σ
 measured for the graphite educt, graphene oxide powders of different batches in the up-scaled 1600 mL reactor and comparable graphene oxide produced by Hummer’s method according to Xu et al. [44].

Sample	Powder Conductivity σ [S/m]
Graphite educt	3.37 × 10^3^
GO—1600 mL batch I	3.89 × 10^2^
GO—1600 mL batch II	3.26 × 10^2^
GO—1600 mL batch III	7.26 × 10^2^
GO—1600 mL batch IV	4.91 × 10^2^
GO—1600 mL batch V	3.96 × 10^2^
GO—1600 mL batch VI	4.83 × 10^2^
GO—Average	4.68 × 10^2^
GO—Hummer’s method [44]	1.56 × 10^−2^

## Data Availability

The data presented in this study are available upon request from the corresponding authors.

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
