# Peer review of "Development and Up-Scaling of Electrochemical Production and Mild Thermal Reduction of Graphene Oxide"

_materials, 2022, doi:10.3390/ma15134639_

Round 1

Reviewer 1 Report

This is an interesting paper providing useful information. Some presentation can be further improved to clarify certain points. Some comments and questions are listed below.

1.     The dimension of Figure a(a) should be provided.

2.     Figure 1. All lines are in the same color. I could hardly see the difference. Could authors possibly improve the presentation of this figure?

3.     Units of equations (2) and (3) should be provided.

4.     From Figure 2. It seems the purity of products is still not good. Any comments on this part? Or alternatively, are there any suggestions for future work?

Author Response

We thank the referee very much for these interesting comments on our manuscript and implemented the comments in our revised manuscript.

Regarding point 4, we agree that the product presents unexfoliated residues. Nonetheless, the residues do not deteriorate the functionalities aimed in polymer-additive application. 

As we propose in the manuscript, the cation (H+, Li+) seems to play a role during exfoliation, suggesting a direction for further investigations to improve the product quality.

Furthermore, the contribution of other parameters like temperature, electrolyte concentration and pretreatment time can be investigated using a two-factorial design evaluation.
Possible ways of purification with a combination of ultrasonication and centrifugation may be considered to further separate the different layer fractions.
In conclusion, we enphasize that the protocol suggested in the manuscript presents an important stepforward in the production of GO as we aim to employ it for our future research. At the same time, we keep in mind the proposed suggestions to continue improving the quality of our products.  

Reviewer 2 Report

In this paper the Authors studied a method for scaling up the production of GO, and discussed

how to produce reduced graphene oxide as a precursor for aeronautical coatings. For that 

purpose, the Authors show how to maintain significant conductive properties, focusing on thermal reduction at lower temperatures for minimal environmental impact. Hence, the main question addressed by the research is how to scale up the production of GO, and how to produce reduced graphene oxide as a precursor for aeronautical coatings.

This study is relevant and interesting, whereas the topic tackled is original.  However, it is not entirely clear what the paper adds to the subject compared with other published material (see papers citred belw, which are absent freom the bibliography). The text is well written, clear and easy to read. The results are supporting the conclusions drawn by the Authors. They are meaningful and original. The conclusions are consistent with the evidence and arguments presented. They address the main question posed. Hence I support publishing the paper. The only recommendation, prior to publication is that the Authors compare their methods, strategy and findings with earleir literature, e.g. 

1) Journal of Materials Research and Technology

Volume 9, Issue 5, September–October 2020, Pages 11587-11610

2) Carbon Trends

Volume 5, October 2021, 100120 Coatings 2021, 11, 297.

Author Response

We thank the referee very much for these interesting comments on our manuscript and the suggested literatures.

We welcomed the suggestion to improve the comparison to other methods, which is now implemented in the manuscript with the paper of Ikram et al., Bocharov et al. and Park et al. at lines 55f., 223f., 304f. and 383ff.

With this we further clarify the scientific contribution of the present manuscript with respect to the environmental impact, in addition to the industrial relevance given by the proposed scalability. 

Reviewer 3 Report

Comments about the paper “Development and up-scaling of electrochemical production and mild thermal reduction of graphene oxide” by Markus Ostermann et al.

The paper reports the preparation of graphene oxide (GO) by an electrochemical procedure and the further reduction of GO by heat treatment at several temperatures to obtain reduced GO (r-GO). The materials are characterized by standard procedures and the conductivity of the samples is measured.

Unfortunately, nothing in this paper is new. There is a large number of papers dealing with the synthesis of GO by electrochemical procedures. Please, see for example the references in the manuscript but there are much more papers in the literature. Similarly, the reduction of r-GO by heat treatment has been widely reported. The allegedly novelty of the work is the so-called “up-scaling” the synthesis procedure. For this task the electrochemical synthesis is carried out in a range between 500 and 1600 ml. It is not realistic to think the synthesis in a 1600 ml vessel can be a reference for industrial purposes.

Therefore, the paper lacks novelty and no new insights are reported.

In my opinion this paper does not deserve publication.

Author Response

We thank the referee very much for this critical comments on our manuscript, which we kindly disagree with.  
Although the field is highly investigated by different groups, the manuscript can contribute to the scientific community as it proves a feasible route to the green up-scalable production of graphene oxide. 

We agree that the up-scaling can be further improved to higher volumes, but it is relevant to show that the reactor volumina have a minor influence on the product quality (linear scaling relationship), as proved by XRD, Raman and XPS measurements. This is not clearly reported in literature for such a system yet. 
In addition, for the vessel size of 1600ml a throughput of about 20~g/h is achieved. This quantity is already sufficient to produce 1~m² of electrothermal De-Icing coating, which is already close to industrial necessary production capacity, given that this process can be run in parallel in a stack design, with a very low footprint. 
We finally emphasize that the production route is relying on environmentally benign chemicals, and the reduction of GO is done at low temperature. This is new and provides an innovative and green production compared to competing process designs in the literature. 
Hence, we have to politely disagree with the referees assessment, and are certain that this is worthy of publication. 

Reviewer 4 Report

The present work shows and describes the up-scaling the electrochemical production of exfoliated graphie material. The paper seems to be interesting and it may add a few new things to the literature. However, before publication the authors should explain two issues:

1. I am not sure, if the synthesized material should be recognized as graphene oxide (GO). As can be seen in Fig. 2a, the diffraction peak which should be signed to GO (around 11 2Theta) has a very small intensity compared to the graphitic peak. In my opinion, during the electrochemical exfoliation graphite was only partially oxidized and partially transformed into GO. Normally, for the GO, only one wide peak around 10 2Theta should be observed, and the peak assigned to the graphite is not present. I suggest to name the sythesized material as partially oxidized graphene material or electrochemically exfoliated graphite.

2. Please explain what is the purpose to compare the conductivity of obtained oxidized graphene materials with the conductivity of exfoliated graphite oxide prepared by Hummers method? In my opinion, it is pretty obvious that prepared materials that are only partially transformed into GO will have much higher conductivity than fully oxidized graphite (GO), because of the lack of graphitic structure (diffraction peak from graphite is absent for GO prepared by Hummers method) and much higher concentration of oxygen functional groups. I think it would be better to compare the conductivity of thermally reduced graphene materials with the reduced GO prepared by Hummers method after its thermal reduction using the same conditions. As the authors themselves indicated in the introduction, the graphene material (reduced graphene oxide) can be used as a filler and coating to enhance the adaptability of the aircraft parts, not GO.

Author Response

We thank the referee very much for these interesting comments on our manuscript. 

1. We agree that the oxidation degree of our product is below the usual value achieved during chemical exfoliation (20 % rather than the usual 40 %). The high oxidation degree leads to an increased interlayer spacing moving the (002) reflex to about 11°. With our lower amount of in-plane oxidation, the interlayer spacing is also increased moving the reflex only slightly to 24-25°.
Consequently, we are keen to name our product electrochemically exfoliated graphene oxide as the names partially oxidized graphene goes in contrast with the relatively high amount of oxygen found with XPS and electrochemically exfoliated graphite goes in contrast with the number of layers obtained out of the XRD measurements. Still, we acknowledge that a real definition of oxidation degree as a function of layer numbers is still missing for graphene oxide materials, making hard to properly name oxidized products in the range between 5 to 25 atomic layers. 

2. We agree that it is reasonable to compare our product to GO produced via Hummers method and then thermally reduced.
Thus, we compared our findings to the work of Bocharov et al. and Park et al. in the thermal reduction section (lines 383ff.)
In the electrical conductivity and XPS measurements section, we are still opting to compare our graphene oxide to the currently used chemical production method and point out the advantage of the mild oxidation during electrochemical exfoliation to highlight the environmental impact. 
In conclusion, we thank the referee for this constructive comment as we found that the conductivity obtained after further thermal reduction is still better with respect to thermally reduced graphene oxide prepared via Hummers methods, up to a temperature of 600 °C. 

Round 2

Reviewer 3 Report

Unfortunately I have the same opinion as before, i.e. the paper lacks novelty and nothing in this paper is new. In my opinion this paper does not deserve publication.

Author Response

We politely dissent from the referee comment, as we agree with the referees 1, 2, and 4 that the work is interesting and worth publication. To prove our point, we would like to list here a series of publications that might help visualize the quality of the paper and answer the questions risen in the first report.

  1. The topic of rGO heat treatment is been improved in the manuscript thanks to suggestions highlighted by other referees as well. Our production method allows to reach a valuable conductivity via mild temperature, making the process more energy-saving and of interest for industries (see the added refs Bocharov et al (2018) and Park et al (2019) at lines 390 -398 of the new version).
  2. About the upscaling, we would like to point out that, to the best of our knowledge, there are no work presenting a feasible upscalable protocol for the system discussed in this manuscript. To make some comparison in literature, Huang et al (2015) have presented a similar approach to the GO production, but remained to the level of proof of concept, not presenting the scalability to different voluminal. Similarly, Tene  et al. (2019) argued to have a system for the large-scale production of GO, but they do not provide a scalability analysis. Moreover, they do not utilize electrochemical exfoliation, but a modified version of the Hummers method. Achee  et al (2018) presented a work about scalability of electrochemically exfoliated graphene which we agree presents a similar approach to the one reported here, but they obtained a maximal production of 5g/h of graphene oxide, that is way lower than the one presented in this manuscript (20 g/h). As a last example, Benoit et al  (2020) presented another interesting work on the scalability, but again here the maximum amount of electrochemical graphene oxide produced is 32 g within 1 day (not better specified), which is still lower than the values reached in this work of 38 g in a complete process of two hours.

The references are now implemented in the manuscript and discussed in the introduction.